# Sex dimorphism in European sea bass (*Dicentrarchus labrax* L.): New insights into sex-related growth patterns during very early life stages

**Sara Faggion**[1], **Marc Vandeputte**[1,2], **Alain Vergnet**[1], **Frédéric Clota**[1,2], **Marie-Odile Blanc**[1], **Pierre Sanchez**[1,2], **François Ruelle**[1], **François Allal**[1]*

**1** MARBEC, CNRS, Ifremer, IRD, Univ Montpellier, Palavas-les-Flots, France, **2** INRAE, AgroParisTech, GABI, Université Paris-Saclay, Jouy-en-Josas, France

* francois.allal@ifremer.fr

**Data Availability Statement:** The datasets underlying our findings are available in the institutional public data repository (SEANOE:

## Abstract

The European sea bass (*Dicentrarchus labrax*) exhibits female-biased sexual size dimorphism (SSD) early in development. New tagging techniques provide the opportunity to monitor individual sex-related growth during the post-larval and juvenile stages. We produced an experimental population through artificial fertilization and followed a rearing-temperature protocol (~16°C from hatching to 112 days post-hatching, dph; ~20°C from 117 to 358 dph) targeting a roughly balanced sex ratio. The fish were tagged with microchips between 61 and 96 dph in five tagging trials of 50 fish each; individual standard length (SL) was recorded through repeated biometric measurements performed between 83 to 110 dph via image analyses. Body weight (BW) was modelled using the traits measured on the digital pictures (i.e. area, perimeter and volume). At 117 dph, the fish were tagged with microtags and regularly measured for SL and BW until 335 dph. The experiment ended at 358 dph with the sexing of the fish. The sex-ratio at the end of the experiment was significantly in favor of the females (65.6% vs. 34.4%). The females were significantly longer and heavier than the males from 103 dph (~30 mm SL, ~0.44 g BW) to 165 dph, but the modeling of the growth curves suggests that differences in size already existed at 83 dph. A significant difference in the daily growth coefficient (DGC) was observed only between 96 and 103 dph, suggesting a physiological or biological change occurring during this period. The female-biased SSD pattern in European sea bass is thus strongly influenced by very early growth differences between sexes, as already shown in previous studies, and in any case long before gonadal sex differentiation has been started, and thus probably before sex has been determined. This leads to the hypothesis that early growth might be a cause rather than a consequence of sex differentiation in sea bass.

https://www.seanoe.org/data/00648/75983/).
Faggion Sara, Vandeputte Marc, Vergnet Alain, Clota Frédéric, Blanc Marie-Odile, Sanchez Pierre, Ruelle François, Allal François (2020). Data on growth during very early life stages of European sea bass (*Dicentrarchus labrax* L.).

**Funding:** The study was funded by the French Ministry of Agriculture (CRECHE2019 project).

**Competing interests:** The authors declare that they have no competing interests.

## Introduction

The phenomenon of sexual size dimorphism (SSD) is common in animal species, and it is represented by the differences in average body size of adult males and females [1]. Female-biased SSD is explained as a situation where females are larger than males, while male-biased SSD is the reverse situation.

Male-biased SSD has been described in various teleost fish species: in different tilapiine strains, adult males are much larger than adult females [2] and have a faster growth rate [3]; the same pattern has been observed in cichlids [4], salmonids [5,6] and catfishes [7]. Male-biased SSD is evolutionary linked to increased male-male competition, territoriality, or female-choice [8]. Conversely, female-biased SSD is linked to increased fecundity of larger females and decreased male-male competition [8]; it has been observed, among the others, in turbot *Psetta maxima* [9,10] and in the European eel *Anguilla anguilla* [11].

Female-biased SSD is also a characteristic of the European sea bass (*Dicentrarchus labrax* L.), one of the major aquaculture species in the Mediterranean area. The females of this species are known to be about 30% heavier than the males from 300–400 g until over 1000 g [12,13]. Furthermore, the common aquaculture practice of size grading has shown that the largest fish selected at 86 days post-hatching (dph) later result to be mostly females [14]. Previous studies exploiting individual tagging suggested that females are already significantly heavier than males from 105 dph (1024 degree days above 10˚C), with a stable 40% difference from 197 to 289 dph [15].

This supports the hypothesis that in European sea bass sex-specific growth may happen before gonadal sex differentiation, as differentiation starts only around 128 dph [14]. In this species, there are no sex chromosomes or "genetic sex", as sex is determined by the combination of multiple genes and environmental temperature (see the review by [16]).

New insights into the onset of sexual dimorphism in European sea bass during the post-larval stages can be gained through the application of ultra-small tagging technologies to individually identify and monitor small-bodied fish. $1 \times 6$ mm RFID glass microtags have been tested in European sea bass, allowing the tracking of individuals from 96 dph (SL, standard length, ~ 36 mm; [15]). More recently, RFID microchip ($0.5 \times 0.5$ mm) tagging has been performed in fish aged 75 dph (SL ~ 20 mm; [17]).

This study exploited the techniques of microchip tagging and microtagging described in previous papers [15,17,18] to identify and follow European sea bass individuals from a post-larval stage (83 dph or 510 degree days above 10˚C) until an age at which a reliable sex identification by gonads inspection was possible (358 dph or 3229 degree days above 10˚C). Growth data were individually recorded during the experiment and related to the sex of the fish as observed at 358 dph. The aim of the study was to identify differences in growth trajectories between males and females at the most precocious stage ever monitored and, in any case, when the morphological sex differentiation of the gonads has not occurred yet.

## Materials and methods

### Production, rearing, and microchip tagging of the experimental fish

The experiment was evaluated by the Ethical Committee (n˚ 036) and authorized by the French Ministry of Higher Education, Research and Innovation (APAFIS#19713-2019010917222576v3). All experimental procedures were conducted following the guidelines for animal experimentation established by the Directive 2010-63-EU of the European Union and the equivalent French legislation.

The production of the experimental fish is detailed in [17]. Briefly, artificial fertilization was performed in March 2019 at the IFREMER experimental facilities (Palavas-les-Flots, France) using the eggs of two dams and the cryopreserved sperm of five sires, from a synthetic F2 line originating from both Atlantic and Mediterranean broodstock. After hatching, larvae were reared in a common garden under controlled conditions (mean rearing temperature: 16.4˚C; salinity: 30.5‰). Fish were tagged with $0.5 \times 0.5$ mm microchips inserted in the peritoneal cavity during five tagging trials between 61 dph (or 372 degree days above 10˚C) and 96 dph (or 596 degree days above 10˚C; 50 fish in each trial; see [17]) and distributed each time to a new tank (N = 250 in total). All surgery was performed under MS-222 (Sigma-Aldrich, 0.07 g/l of seawater) anesthesia, and all efforts were made to minimize suffering during the manipulation.

The conditions of salinity and temperature of all the different tanks were the same as the common garden tank and were maintained until 112 dph (or 720 degree days above 10˚C); the following 5 days the temperature was gradually increased to 20˚C. Fish were then reared at a mean temperature of 20.3˚C (19.3–21.2˚C), salinity of 36.5‰ and photoperiod at 12L:12D (light:dark).

## Growth monitoring, microtagging and sex recording

Biometric measurements were performed at 83, 89, 96, 103, 110, 117, 137, 150, 165, 180, 201, 223, 265, 302 and 335 dph. Each fish was anesthetized with MS-222 (Sigma-Aldrich, 0.07 g/l of seawater; [19]), recognized through microchip ID reading, then placed over a light table (Ultra Slim Light Box, Microlight) and photographed using a stand with a digital camera (12.2 megapixel). The measure of the standard length of each fish were performed through image analysis (ImageJ software 1.51; [20]; see Supplementary material 1, S1 Fig).

When fish reached 117 dph (or 756 degree days above 10˚C), they were all weighed and tagged with a second tag through intra-coelomic implantation of $1 \times 6$ mm glass microtags (Lutronics, Nonatec RFID, Lutronics International, Rodange, Luxembourg) following the protocol by [18]. This was done to prevent loss of the identity due to the increasing difficulties of optical microchip reading as fish grow, and thus to enable individual growth data recording and correct sex assignment at the end of the experiment. The fish were anesthesized using 32.5 μl from a 10% stock solution of ethyl-p-aminobenzoate (Benzocaine E1501, Sigma-Aldrich) dissolved in 100% ethanol, per 100 ml of seawater solution. The tagging protocol consisted in piercing a hole in the abdominal cavity of the fish with an 18-gauge needle, the microtag was picked up with a Dumont n˚ 3 forceps, inserted and pushed inside into the abdominal cavity through the hole. All efforts were made to minimize animal suffering. The fish were then transferred to a tank of isosmotic 0.2 μm filtered and sterilized seawater for recovery (to avoid osmotic stress and prevent infections) and they were allowed to rest for 1 to 2 h before being returned to their rearing tank. Fish were reared in a common garden tank from 117 to 358 dph with the same conditions described above (mean temperature of 20.28˚C, 36.5‰ salinity and photoperiod at 12L:12D).

During the biometric measurement performed between 137 and 335 dph, fish were anesthetized as described above, adjusting the anesthetic solution of ethyl-p-aminobenzoate and seawater according to the increasing size of the fish, the microtag was read, the body weight and the standard length were individually registered.

At 358 dph (or 3229 degree days above 10˚C) pre-anesthetized fish were euthanized with an excess of benzocaine solution, taking care to avoid any animal suffering during the fishing and the placing of the animals inside the benzocaine bath. The sex was identified macroscopically through the direct observation of the gonads or using a gonadal squash [21] when the visual observation was ambiguous (see Supplementary material 2, S2 Fig).

## Prediction of body weight from digital picture measurements and prediction of standard length

The biometric measures performed on early stage fish (83 to 110 dph) relied on image analyses that allowed the measurement of length, height, perimeter and area. To build a model to estimate the body weight from digital picture measures, we followed the procedure detailed by [22]. During the biometric measurements performed at 83, 89, 96, 103 and 110 dph, 50 additional fish from the stock rearing tank were randomly chosen and sacrificed with an excess of anesthetic (MS-222) to directly measure the length and the weight of each fish (total number of fish = 250). The standard length was obtained with a V-12B 12" vertical optical comparator (Nikon) that allowed an accurate measure through magnification of the larva. The measure of the body weight was achieved using a precision scale (to the nearest 0.01 g) after drying the fish with absorbent paper.

In addition, a digital picture of each fish was taken following the same procedure used for the experimental fish. ImageJ software 1.51 [20] was used to perform image analysis obtaining the measures of area, perimeter, length, and height. The steps of image analysis are fully described in [22].

A volume index was calculated for each fish from height and length as:

$$Volume = \frac{\pi * Height^2 * Length}{12}$$

Pearson's coefficient of correlation (r) between measurements obtained from image analysis and measurements obtained directly was estimated in R using cor.test function (package *stats*, R version 3.5.0, [23]). Multiple regression models using length, height, perimeter, area and volume were tested using lm and glm functions in R (package *stats*). The efficiency of the models and regression equations exploiting different combinations of the traits to predict BW was evaluated through the coefficient of determination ($R^2$) and the Akaike information criterion (AIC). The validation was performed as described in [22] dividing the dataset into a "model set" (74% of the dataset) and a "validation set" (26% of the dataset; 13 randomly chosen fish for each biometric measurement).

The r between estimated and measured BW as estimated to assess the accuracy of the prediction model.

During the biometric measurement performed at 117 dph, only BW was directly measured on the fish; for this reason, a model to estimate standard length using body weight was built. The data from the 50 additional fish sacrificed during each biometric measurement was used; standard length and body weight were log-transformed. The procedures followed were the same as the model built for body weight.

The predictive models were then applied to the experimental fish dataset to estimate the body weight of fish aged 83, 89, 96, 103 and 110 dph and the standard length of fish aged 117 dph.

## Modeling and comparison of growth curves for BW and SL

Growth curves for BW and SL were built through a linear regression of $BW^{1/3}$ and SL on degree days above 10˚C (d.days). Data were divided into three growth periods: from 83 to 117 dph (510 to 756 degree days), from 137 to 223 dph (756 to 1838 degree days), and from 265 to 335 dph (2271 to 2992 degree days). Growth curves were built separately for each growth period. Growth curves were then compared through ANCOVA with the model:

$$Y_{ij} = m_u + sex_i + a(d.days) + b_i(d.days)$$

where $m_u$ is the intercept, $sex_i$ is the effect of sex, $a$ is the general slope, $b_i$ the sex-specific slope. The significance threshold for the statistical tests was p-value < .05. The tests were performed in R version 3.5.0, package *stats* [23] using the functions lm and anova.

### Fulton's K condition factor and daily growth coefficient

Fulton's body condition factor K [24] is a non-lethal morphometric index estimate of body condition, which states that heavier fish of a given length are in better condition. It was estimated as:

$$K = \frac{100,000(BW)}{SL^3}$$

The daily growth coefficient was computed from the body weight data for each period between two biometric measurements. The formula was the following:

$$DGC = \frac{\omega_f^{\frac{1}{3}} - \omega_i^{\frac{1}{3}}}{t} \times 100$$

where $\omega_f$ is the final measure for body weight, $\omega_i$ is the initial measure for body weight and $t$ is the number of days.

### Statistical analyses

The number of males and females in the population were compared through $\chi^2$ tests.

Data for SL, BW, DGC, and K were checked for normality and for homoscedasticity through Shapiro-Wilk and Bartlett's tests. When the assumptions of normality and homoscedasticity were respected, data were compared through ANOVA to check sex-related early growth patterns. Post-hoc analyses to adjust the p-values were performed through Tukey's test. When data were assessed as non-normal and/or variances were not homogeneous, non-parametric Wilcoxon-Mann-Whitney test was performed (one test at once; the p-value was adjusted through Bonferroni correction). The significance threshold for the statistical tests was p-value < .05. All the tests were performed in R version 3.5.0, package *stats* [23].

## Results

### Prediction of body weight from digital picture measurements and prediction of standard length

Pearson's coefficient of correlation (r) between measurements obtained from image analysis and measurements obtained directly were all high and significant, ranging from 0.9533 to 0.9963 (see S1 Table in Supplementary material 3 for details). The traits with the greatest correlation with BW were area (0.9898, *p*-value < .0001) and volume (0.9963, *p*-value < .0001).

The model exploiting area, perimeter and volume was the one with the lowest AIC (-1053.5) and the greatest $R^2$ (0.9942; see S2 Table in Supplementary material 3 for further details). The model was the following:

$$BW = 0.0153 + 0.011(Area) - 0.0030(Perimeter) + 0.0008(Volume) \tag{1}$$

The coefficient of correlation between measured and estimated BW using model (1) was 0.9966 (*p*-value < .0001). The BW in the "model set" was estimated with a $R^2$ of 0.9942 (r between measured and estimated BW of 0.9971, *p*-value < .0001), in the "validation set" with a $R^2$ of 0.9906 (r between measured and estimated BW 0.9953, *p*-value < .0001).

The coefficient of correlation between the logarithm of the measured SL and the logarithm of the measured BW was significantly high (0.9873, $p$-value $< .0001$). The logarithm of SL was estimated for the fish aged 117 dph (when only BW was directly measured) with the following model, having an AIC equal to -1111.5 and an $R^2$ equal to 0.9708:

$$\log(SL) = 0.7350 + 0.2838(\log(BW)) \tag{2}$$

The coefficient of correlation between measured and estimated SL using model (2) was 0.9872 ($p$-value $< .0001$). The SL in the "model set" was estimated with a $R^2$ of 0.9708 (r between measured and estimated SL of 0.9853, $p$-value $< .0001$), in the "validation set" with a $R^2$ of 0.9814 (r between measured and estimated SL 0.9907, $p$-value $< .0001$). For further details, see S3–S5 Figs in Supplementary material 3.

## Proportions of males and females and sex-related growth patterns

The reliable identification of the sex was possible, either through visual observation of the gonads or gonadal squash, for the 98.4% of the fish; for the remaining 1.6%, the gonadal differentiation was not completed yet, entailing some degree of uncertainty in the assignment of the sex. These fish were then removed from the dataset. Globally, at the end of the experiment, 87 females and 45 males were detected, with a sex-ratio in favor of the females of 65.9% vs. 34.1% for males (Tables 1 and 2), which was significantly different ($\chi^2 = 13.364$, $p$-value $= 3 \times 10^{-4}$).

Four outlier fish displaying inconsistent SL and BW measurements at 103 and 110 dph were removed from the analysis.

Differences in terms of growth patterns were observed between females and males (Table 1; Figs 1 and 2). On average, females were longer compared to males from 103 dph, when females were 6% longer than males, and a significant difference was maintained until 165 dph, with females close to 4% longer than males. From 180 dph until the end of the experiment, the difference in length between females and males was small (around 2.5% in favor of females), and not significant.

Table 1. Body weight, standard length, and Fulton's body condition factor K.

| Age (dph) | N (%) | | SL (mm) ± SD | | BW (g) ± SD | | Fulton's K ± SD | |
|---|---|---|---|---|---|---|---|---|
| | F | M | F | M | F | M | F | M |
| 83 | 11 (61.1%) | 7 (38.9%) | 23.3 ± 1.3 | 22.8 ± 2.2 | 0.17 ± 0.04 | 0.15 ± 0.04 | 1.30 ± 0.11 | 1.23 ± 0.10 |
| 89 | 21 (70.0%) | 9 (30.0%) | 25.9 ± 1.9 | 25.2 ± 2.5 | 0.25 ± 0.06 | 0.22 ± 0.06 | 1.40 ± 0.11 | 1.35 ± 0.09 |
| 96 | 28 (62.2%) | 17 (37.8%) | 28.2 ± 1.7 | 26.7 ± 2.9 | 0.34 ± 0.08 | 0.30 ± 0.10 | 1.49 ± 0.10 | 1.45 ± 0.07 |
| 103 | 41 (60.3%) | 27 (39.7%) | 31.0 ± 2.0** | 29.4 ± 3.0** | 0.47 ± 0.10** | 0.39 ± 0.13** | 1.54 ± 0.09* | 1.48 ± 0.10* |
| 110 | 36 (56.5%) | 26 (41.9%) | 33.5 ± 2.3* | 32.2 ± 2.9* | 0.62 ± 0.14 | 0.55 ± 0.15 | 1.61 ± 0.09 | 1.62 ± 0.09 |
| 117 | 87 (65.9%) | 45 (34.1%) | 35.7 ± 2.2* | 34.5 ± 2.8* | 0.78 ± 0.17* | 0.70 ± 0.19* | 1.67 ± 0.05* | 1.64 ± 0.07* |
| 137 | 87 (65.9%) | 45 (34.1%) | 57.3 ± 3.7* | 55.3 ± 5.2* | 2.38 ± 0.44* | 2.16 ± 0.52* | 1.25 ± 0.09 | 1.26 ± 0.11 |
| 150 | 87 (65.9%) | 45 (34.1%) | 62.2 ± 4.3 | 60.6 ± 6.1 | 3.21 ± 0.62* | 2.93 ± 0.78* | 1.32 ± 0.11 | 1.29 ± 0.13 |
| 165 | 87 (65.9%) | 45 (34.1%) | 74.4 ± 5.0* | 71.7 ± 6.9* | 5.35 ± 1.06* | 4.89 ± 1.36* | 1.29 ± 0.12 | 1.30 ± 0.16 |
| 180 | 87 (65.9%) | 45 (34.1%) | 83.7 ± 5.4 | 81.2 ± 7.8 | 8.26 ± 1.73 | 7.63 ± 2.18 | 1.39 ± 0.08 | 1.39 ± 0.07 |
| 201 | 87 (65.9%) | 45 (34.1%) | 95.1 ± 8.5 | 93.7 ± 11.1 | 14.27 ± 3.14 | 13.19 ± 4.08 | 1.65 ± 0.22* | 1.57 ± 0.21* |
| 223 | 87 (65.9%) | 45 (34.1%) | 118.8 ± 8.2 | 115.9 ± 11.0 | 23.17 ± 5.18 | 21.55 ± 6.46 | 1.36 ± 0.08 | 1.35 ± 0.08 |
| 265 | 87 (65.9%) | 45 (34.1%) | 142.3 ± 10.3 | 138.3 ± 14.2 | 40.97 ± 9.44* | 37.58 ± 12.20* | 1.40 ± 0.09 | 1.38 ± 0.10 |
| 302 | 86 (65.6%) | 45 (34.4%) | 161.7 ± 11.8 | 157.8 ± 16.6 | 59.71 ± 14.00 | 55.11 ± 18.50 | 1.39 ± 0.08* | 1.35 ± 0.11* |
| 335 | 86 (65.6%) | 45 (34.4%) | 176.6 ± 12.9 | 172.0 ± 17.5 | 79.19 ± 18.95 | 73.44 ± 24.76 | 1.41 ± 0.09 | 1.39 ± 0.09 |

Number and percentage of males and females (N%), mean standard length (SL, mm) ± standard deviation (SD), mean body weight (BW, g) ± standard deviation and Fulton's K condition factor ± standard deviation for each age. Asterisks indicate significant differences between males and females ($p$-value $< .01$ **; $p$-value $< .05$ *).

**Table 2. Daily growth coefficient.**

| Age interval (dph) | N (%) | | DGC ± SD | |
|---|---|---|---|---|
| | F | M | F | M |
| 83–335 | 11 (61.1%) | 7 (38.9%) | 1.44 ± 0.14 | 1.38 ± 0.17 |
| 83–89 | 8 (53.3%) | 7 (47.7%) | 0.91 ± 0.37 | 0.92 ± 0.15 |
| 89–96 | 15 (71.4%) | 6 (28.6%) | 1.02 ± 0.20 | 0.89 ± 0.23 |
| 96–103 | 22 (59.5%) | 15 (40.5%) | 1.02 ± 0.26** | 0.81 ± 0.22** |
| 103–110 | 26 (54.2%) | 22 (45.8%) | 1.15 ± 0.25 | 1.20 ± 0.17 |
| 110–117 | 34 (56.7%) | 26 (43.3%) | 0.96 ± 0.39 | 0.86 ± 0.30 |
| 117–137 | 87 (65.9%) | 45 (34.1%) | 2.08 ± 0.17 | 2.03 ± 0.18 |
| 137–150 | 87 (65.9%) | 45 (34.1%) | 1.06 ± 0.25 | 1.05 ± 0.25 |
| 150–165 | 87 (65.9%) | 45 (34.1%) | 1.82 ± 0.23 | 1.75 ± 0.25 |
| 165–180 | 87 (65.9%) | 45 (34.1%) | 1.80 ± 0.36 | 1.79 ± 0.61 |
| 180–201 | 87 (65.9%) | 45 (34.1%) | 1.91 ± 0.31 | 1.84 ± 0.56 |
| 201–223 | 87 (65.9%) | 45 (34.1%) | 1.92 ± 0.27 | 1.90 ± 0.27 |
| 223–265 | 87 (65.9%) | 45 (34.1%) | 1.41 ± 0.23 | 1.32 ± 0.28 |
| 265–302 | 87 (65.9%) | 45 (34.1%) | 1.24 ± 0.23 | 1.21 ± 0.27 |
| 302–335 | 86 (65.6%) | 45 (34.4%) | 1.16 ± 0.29 | 1.15 ± 0.27 |

Number and percentage of males and females (N%), mean daily growth coefficient (DGC) ± standard deviation for each age interval. Asterisks indicate significant differences between males and females ($p$-value < .01 **).

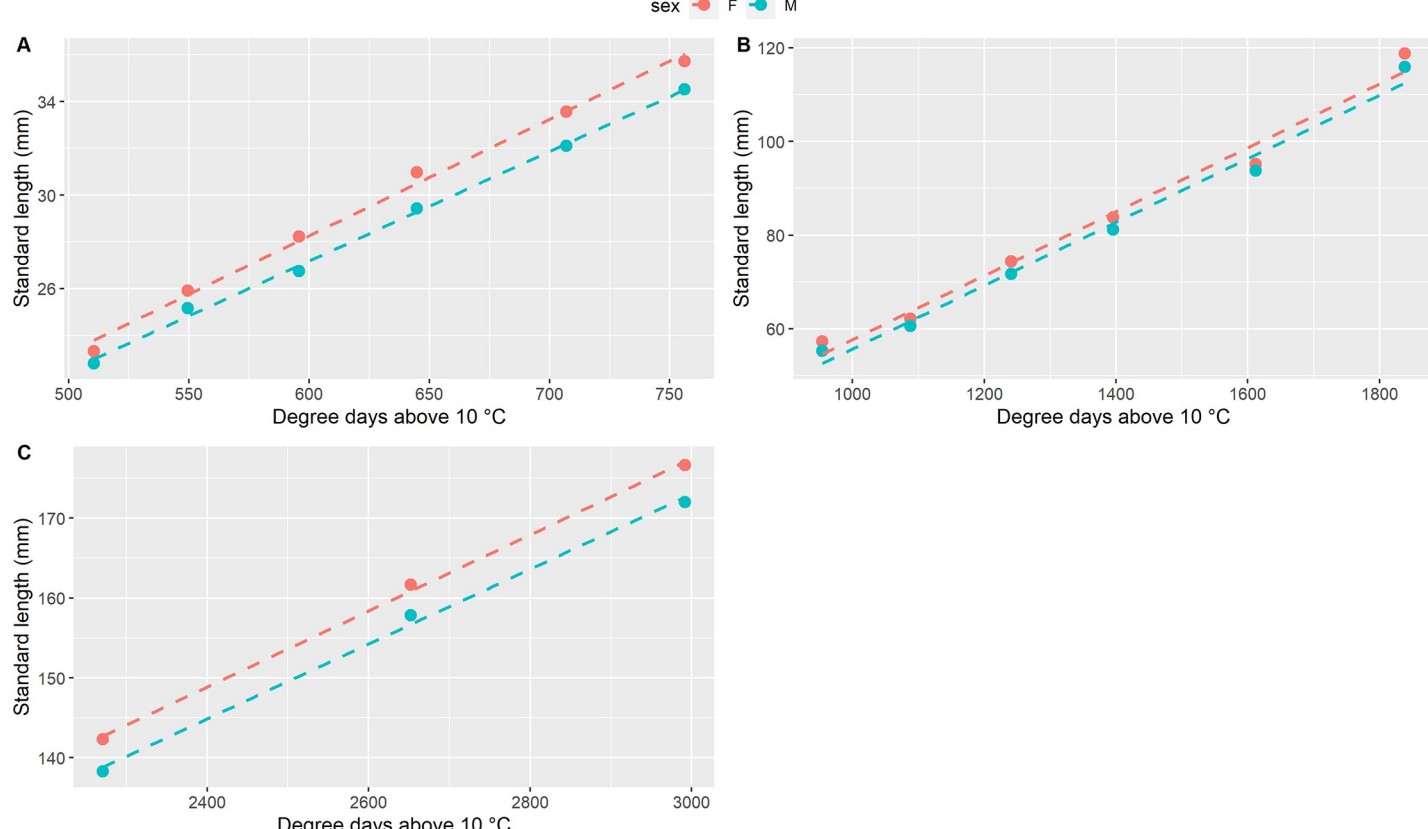

**Fig 1. Growth curve for standard length.** Standard length growth curve for females (F) and males (M) in the three growth periods: A) from 83 to 117 dph (510 to 756 degree days); B) from 137 to 223 dph (756 to 1838 degree days); C) from 265 to 335 dph (2271 to 2992 degree days).

The comparison of the growth curves built for standard length (Fig 1) evidenced a significant effect of sex on the intercept during the first growth period (from 83 to 117 dph; $p$-value = $2.57 \times 10^{-4}$). The general slope of SL on degree-days ($a$) was significant ($p$-value = $0.07 \times 10^{-9}$), whereas the sex-specific slope $b_i$ was not significant ($p$-value = 0.20). For the following periods (137 to 223 dph and 265 to 335 dph), only the general slope was significant ($p$-values = $0.14 \times 10^{-7}$ and $1.28 \times 10^{-3}$, respectively), while the effects of sex and the sex-specific slope $b_i$ were not significant (for $sex_i$: $p$-values = 0.26 and 0.05, respectively; for $b_i$: $p$-values = 0.93 and 0.85, respectively).

Body weight followed approximately the same pattern (Table 1): females were on average heavier than males at 103 dph (females were 20.5% heavier than males), between 117 and 165 dph (females were about 10% heavier than males), and at 265 dph. From 180 dph until the end of the experiment, the difference in weight between females and males was stable, and close to 8% in favor of females, although not significant most of the time (except at 265 dph).

During the first three biometric measurements (at 83, 89 and 96 dph), even though the differences were not significant, females were already around 2 to 6% longer and 13% heavier than males.

The comparison of the growth curves built for body weight (Fig 2) followed a similar pattern as standard length, with a significant effect of sex on the intercept during the first growth period (83 to 117 dph; $p$-value = $3.20 \times 10^{-4}$), a significant effect of general slope ($p$-value = $0.04 \times 10^{-9}$) and a non-significant sex-specific slope ($p$-value = 0.45). For the growth period

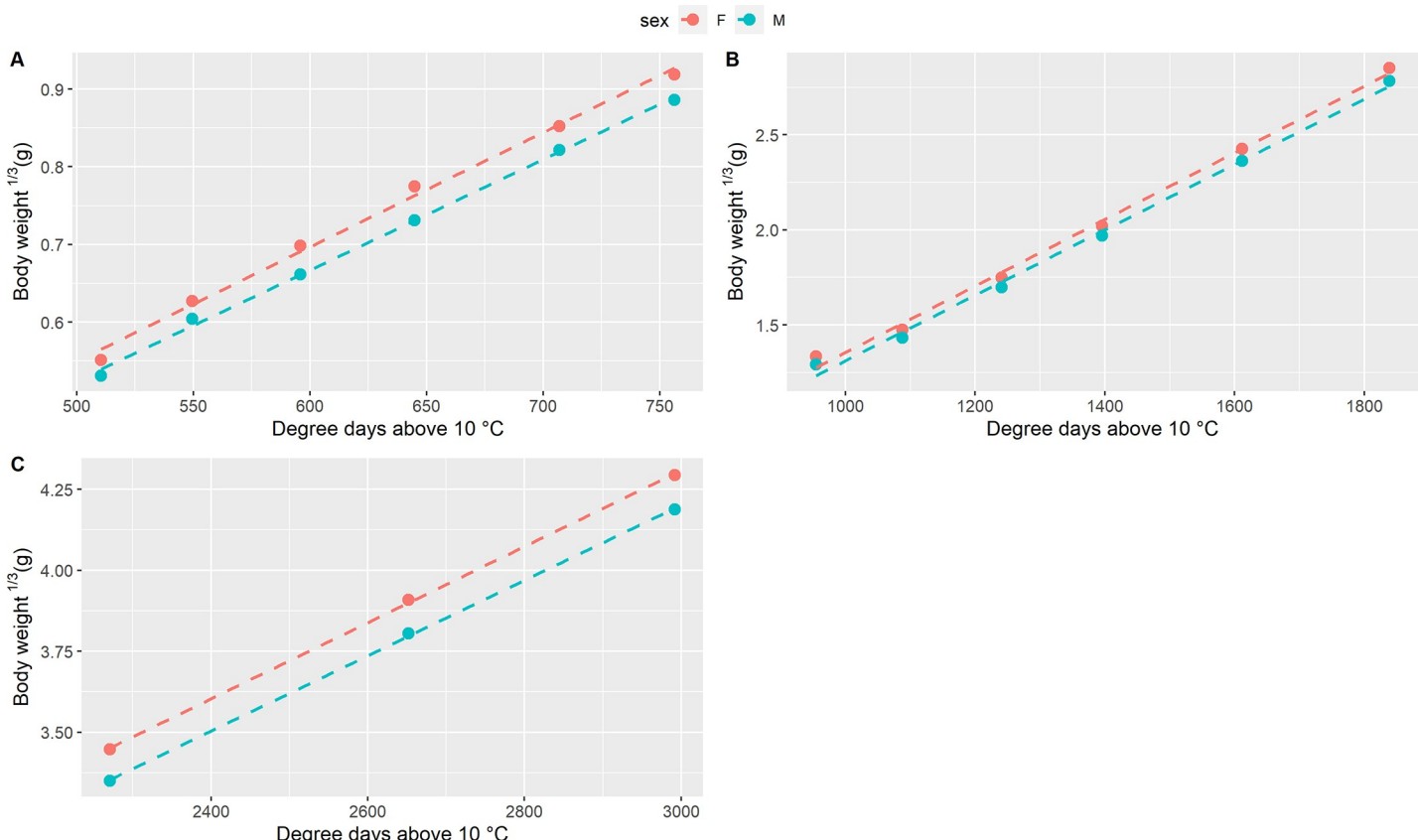

**Fig 2. Growth curve for body weight.** Body weight growth curve for females (F) and males (M) in the three growth periods: A) from 83 to 117 dph (510 to 756 degree days); B) from 137 to 223 dph (756 to 1838 degree days); C) from 265 to 335 dph (2271 to 2992 degree days).

between 137 and 223 dph, only the general slope $a$ was significant ($p$-value = $0.08 \times 10^{-9}$), while the effect of sex and the sex-specific slope $b_i$ were not significant ($p$-values = 0.06 and 0.71, respectively).

During the last growth period (265 to 335 dph), the effect of sex became significant again ($p$-value = $0.75 \times 10^{-2}$), and, as seen before, the general slope $a$ was significant ($p$-value = $0.17 \times 10^{-3}$) while the sex-specific slope $b_i$ was not significant ($p$-value = 0.72)

### Fulton's K condition factor and daily growth coefficient

The females were in better body conditions compared to males in most of the biometric measurements, with significant differences of Fulton's K condition factor at 103, 117, 201 and 302 dph (Table 1). At 110, 137, 165 and 180 dph, the Fulton's K condition factor was comparable between females and males.

The daily growth coefficient based on body weight (DGC) was higher in females in almost all the periods analyzed, with the only exception of the interval between 103 and 110 dph (Table 2). Significant differences between males and females were detected only during the interval between 96 and 103 dph, where the DGC of females was 26% higher than that of males.

## Discussion

The miniaturization of fish tagging technologies has enabled the identification and tracking of individuals from an early life stage, providing the opportunity of studying many biological and physiological changes occurring during these sensitive phases.

Recent papers claimed the effectiveness of microtags [15] and microchips [17] as tagging tools for European sea bass post-larval individuals. In this study, we used a combination of these two tagging methods to efficiently identify the fish during the post-larval stage with microchips (from a mean SL of ~23 mm and an age of 83 dph to a mean SL of ~33 mm and 110 dph) and during the juvenile stage with microtags (from a mean SL of ~36 mm and an age of 117 dph to a mean SL of ~171 mm and an age of 358 dph). This allowed us to record individual growth data through repeated biometric measurements. At the end of the experiment, the individual growth data were related to the sex to gain knowledge about early sex-related growth patterns in the European sea bass.

Our study confirmed and strengthened the already known sex dimorphic growth pattern in the European sea bass [12–15], providing evidence of significant SSD in favor of the females in terms of body weight and standard length. While the pattern of SSD after 10 months of age is well known, with a maximal difference at ~1 year of age, followed by a slow decay [13,25], its earlier dynamics remained poorly described, due to the inability to tag fish before SSD builds up. The earliest tagging study to date was that of [15], which showed that a 31% SSD for weight in favor of females was already established at a size of 0.59 g mean weight and 27–53 mm total length. Other experiments with size graded groups have shown that SSD is already established at a size of 36–45 mm total length as sorting the largest individuals at that size resulted in a clear excess of females, compared with the general population [14].

In the present study, we started measuring growth on the fish 22 days before [15] (83 vs 105 dph), but the difference in terms of developmental stages was even greater, as [15] used a rearing protocol more similar to hatcheries standard procedures, where temperature is raised from 16.5 to 22°C an earlier date (60 dph, Chatain, pers. comm, vs. 112 dph in our study). We were thus able to individually follow the growth of future males and females starting from 0.16 g instead of their 0.59 g.

At 83 dph (510 degree days above 10˚C), females were already heavier and longer compared to the males, but the direct comparison was not statistically significant. From 30.35 mm mean SL and 0.44 mean BW (103 dph, 645 degree days above 10˚C) the differences between males and females became significant, until 73.15 mm mean SL and 5.12 mean BW (165 dph, 1241 degree days above 10˚C), joined with general better conditions of females, measured as Fulton's K condition factor. Nevertheless, the modeling of the growth curves for standard length and body weight and the significantly different intercepts for females and the males, suggested that significant differences in size between sexes may exist since 83 dph, but were probably not revealed due to the limited samples size at those ages. Indeed, the sample size during the first period (83 to 110 dph) was rather low, since fish were not all tagged at the beginning of the experiment but at different ages, as we did not initially know if they would survive and grow normally after such an early tagging (see [17] for details). Some difficulties linked to the optical reading of the microchip, especially at ~ 33 mm mean SL (110 dph), also made that not all fish had a complete set of growth measurements. At 265 dph, the sex effect on body weight (~ 39 g mean BW) became again significant.

Taking the three growth periods (83–117, 137–223 and 265–335 dph) as a whole, the growth rates of the females and the males were similar, but during specific moments the comparison of the growth rates, measured as DGC, was statistically significant, as it was detected from ~ 27 mm to ~ 30 mm SL (96 to 103 dph), when the DGC in females was 26% higher than in males. Thus, apart from the possible difference in initial size, the main period for the onset of SSD in sea bass seems to start around 0.33 g mean BW and 27.5 mm mean SL (96 dph, 596 degree-days above 10˚C). This period is also an important period for sex determination in sea bass, as rearing fish at a cold temperature ($< 17˚C$) beyond that time orients sex determination towards males, while earlier cold rearing promotes female sex determination [26]. In any case, this phase of faster growth of females occurs well before the start of histological sex differentiation, which occurs first in females, at a SL of 80–100 mm [14,27,28]. The first signs of molecular sex differentiation (higher expression of aromatase *cyp191a1* in future females) are observed somewhat earlier, at a SL of 55 mm [29], but this still happens much later than the onset of differential growth, which started around 27.5 mm SL in our experiment. This leads to two non-exclusive hypotheses. Firstly, it may be that the differentiation pathway between males and females starts earlier than what has been evidenced for the moment. Microchips could help the study of this in the future, giving access to the future sex of fish as small as 23 mm, but this would have to be coupled with non-lethal sampling for gene expression, which is far from simple at such a small size. Secondly, we should consider the possibility that faster growth would be the cause and not the consequence of sex differentiation towards females. Indeed, when molecular differentiation appears, at 55 mm in [29] or at 96 dph in [26], sex cannot be considered fully determined yet, as the sex of individuals can still be influenced by hormonal treatments [30] or by temperature [26]. It could be influenced by growth, following the threshold model for sex determination proposed by [31], under which at a critical time in development, a sexually undifferentiated gonad will develop as an ovary or as a testis depending on whether it has attained a certain size above or below a threshold. Modifying growth to modify sex has already been tested before, but with larger fish. It was shown that manipulating growth by food restriction starting at 80 or 40 mm SL did not impact sex-ratios in the treated groups [32]. Considering that we showed differential growth at 27.5 mm (96 dph) and that the analysis of growth curves even shows it may have pre-existed at 23 mm (83 dph), the hypothesis that very early growth (from 20 to 40 mm SL) growth may be a cause of female sex differentiation in sea bass should be considered for further research.

We observed a lower SSD between males and females compared to previous studies [13,15,25] which could be linked to the fact that long cold rearing temperature also tends to

decrease SSD in European sea bass [26]. This may also have been influenced by the population used, which is a mixed population between Atlantic and Mediterranean sea bass. There are important differences in growth dynamics between these two lineages [33], although population differences in SSD have not been investigated for the moment.

Another aspect that may have affected SSD in the present study is the effect of microchip tagging on the fish. Fish were not sorted before microchip tagging, but they were randomly chosen from a common garden tank and tagged [17]. Anyway, the manipulation itself could have acted as a sorting event, selecting, *de facto*, the "stronger" fish characterized by a greater growth potential, and thus eliminating the smaller fish, more likely to develop as males, which may at the same time decrease SSD if the smallest males are removed from the population, and increase the proportion of females in fish surviving until sexing.

In a previous paper [17], it was reported that microchip tagging did not affect survival rates nor growth rates of the tagged group in comparison with the untagged group. Analyzing the sex ratio of the tagged group (i.e. the fish of the current experiment) and the untagged group (i.e. the control group in [17]), we observed 65.6% of females and 34.4% of males in the first case, with a proportion of 1.91 females per male; 54.9% of females and 45.1% of males in the second case, with a proportion of 1.22 females per male. The proportion of males and females was not statistically different between groups ($\chi^2 = 1.96$, *p*-value = 0.16), suggesting that the microchip tagging was not the cause of the unbalanced sex ratio we observed in our experiment.

This is also in contrast with the common observations of strongly unbalanced sex-ratio in favor of males in cultured sea bass; indeed, the standard hatchery practices imply high rearing temperatures, that play a role in the masculinization of developing fish [14,30]. In our case, we followed a particular rearing-temperature protocol to obtain a roughly balanced sex ratio. The experimental fish were exposed to low rearing temperatures (~16˚C) during the first part of their life (from hatching to approximately 33 mm of SL, corresponding in our fish to 112 dph), and to higher temperatures (~20˚C) during the second part of their life, targeting a balanced sex-ratio, following [34]. However, recent results show that continuing exposure to cold temperature is likely to have an opposite effect on sex determination, progressively favoring males with time spent below 17˚C beyond 55–75 dph [26]. This may indirectly support our previous hypothesis that tagging may have indirectly increased the proportion of females. However, it has to be noted that the variation of sex-ratios in sea bass in different experiments using the same temperature treatment remains very high, for reasons that are not identified for the moment [26,34].

The fact that SSD in sea bass is established very early had already been evidenced indirectly by sorting experiments [14,28,32], and using genetic links by repeatedly sampling the same families at different ages [25]. This is more precisely documented by the present experiment, by monitoring the individual growth of future males and females starting at 23 mm standard length, at 83 dph. For the first time, we could identify a stage at which differential growth happens between 96 and 103 dph (596 to 645 degree days above 10˚C, 27.5 to 30.3 mm SL, 0.32 to 0.44 g BW), as well as a possible SSD in favour of female already existing at 83 dph. This provides key information to study the hypothesis that faster growth may cause female differentiation in this species, which is plausible as SSD seems to establish before the first known signs of sex differentiation.

## Supporting information

**S1 Fig. Image analysis to recover length, height, perimeter and area of the fish.**
(PDF)

**S2 Fig. Determination of the sex of the fish.**
(PDF)

**S3 Fig. Regression between measured and estimated values of fish body weight using the model (1) for the "model set" (blue circles) and the "validation set" (red triangles).**
(PDF)

**S4 Fig. Regression between logarithm of the measured standard length and logarithm of the measured body weight.**
(PDF)

**S5 Fig. Regression between logarithm of measured and estimated values of fish standard length using the model (2) for the "model set" (blue circles) and the "validation set" (red triangles).**
(PDF)

**S1 Table. Prediction of body weight from digital picture measurements and prediction of standard length.**
(PDF)

**S2 Table. Coefficient of determination ($R^2$) and Akaike information criterion (AIC) for models of multiple regression to estimate body weight.**
(PDF)

## Acknowledgments

We wish to thank Intellibio (Seichamps, France) for providing technical support and instrumentation.

## Author Contributions

**Conceptualization:** Marc Vandeputte, François Allal.

**Data curation:** Sara Faggion.

**Formal analysis:** Sara Faggion.

**Funding acquisition:** Marc Vandeputte, François Allal.

**Methodology:** Sara Faggion.

**Resources:** Sara Faggion, Alain Vergnet, Frédéric Clota, Marie-Odile Blanc, Pierre Sanchez, François Ruelle.

**Supervision:** Marc Vandeputte, François Allal.

**Visualization:** Sara Faggion.

**Writing – original draft:** Sara Faggion.

**Writing – review & editing:** Marc Vandeputte, François Allal.

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
