## [Decision Letter · Decision Letter 0]

18 Nov 2020

PONE-D-20-28397

Sex dimorphism in European sea bass (Dicentrarchus labrax L.): new insights into sex-related growth patterns during very early life stages

PLOS ONE

Dear Dr. Allal,

Thank you for submitting your manuscript to PLOS ONE. After careful consideration, we feel that it has merit but does not fully meet PLOS ONE’s publication criteria as it currently stands. Therefore, we invite you to submit a revised version of the manuscript that addresses the points raised by reviewers..

We look forward to receiving your revised manuscript.

Kind regards,

Hanping Wang

Academic Editor

PLOS ONE

2. In your Methods section, please include a comment about the state of the animals following this research. Were they euthanized or housed for use in further research?

If any animals were sacrificed by the authors, please include the method of euthanasia and describe any efforts that were undertaken to reduce animal suffering.

3. Please include a copy of Table 3 which you refer to in your text on page 8.

**Comments to the Author**

1. Is the manuscript technically sound, and do the data support the conclusions?

Reviewer #1: Partly

Reviewer #2: Partly

2. Has the statistical analysis been performed appropriately and rigorously? 

Reviewer #1: No

Reviewer #2: Yes

3. Have the authors made all data underlying the findings in their manuscript fully available?

Reviewer #1: Yes

Reviewer #2: Yes

4. Is the manuscript presented in an intelligible fashion and written in standard English?

Reviewer #1: Yes

Reviewer #2: Yes

5. Review Comments to the Author

Reviewer #1: Please see uploaded pdf file

Reviewer #2: General comments

The work entitled “Sex dimorphism in European sea bass (Dicentrarchus labrax L.): new insights into sex-related growth patterns during very early life stages” studied sexual size dimorphism in European sea bass. There are three main concerns about the MS:

(1) The main weakness of the MS is lacking of histological observation of sex differentiation progress. Sex differentiation in fish is related to age (days post-fertilization), rearing conditions (e.g. temperature, nutrition), growth rate, and body weight/length. Without gonadal histology will make the results to be uncertain.

(2) There are usually three aspects of sex differentiation, cellular, morphological, and molecular sex differentiation. Molecular sex differentiation is much earlier than cellular and morphological sex differentiation, and is the cause of the other two. However, it is usually difficult to detect molecular sex differentiation until a late stage, because of the sampling problem of undifferentiated or differentiating gonad. Therefore, without the detect of molecular sex differentiation or morphological sex differentiation, the main conclusion “early growth may be a cause rather than a consequence of sex determination in sea bass” is unreliable.

(3) Even though authors make progress in the SSD research using microchip tagging, the tagging process is also the main affecting factor of the results as authors discussed. Eliminating small sized fish during sorting for microchip tagging artificially altered the sex ratio of the experimental population. The sorting process may artificially advance and speed up the SSD because fast-growing females were dominating the population. In addition, as observed in the work, eliminating the most slow-growing male repressed the SSD. It may be the cause why the SSD was disappear in the late stage which is in contrast with previous reports.

Please also find some specific comments as follows.

Specific comments

1. Line 28: The term “sex determination” is misused here and thereafter in the following context.

2. It should be SSD, not SDD.

3. Are there replicates for the experiments? It had not mentioned in the MS.

3. Discussion section. The citation about an unpublished work again and again is not appropriate.

6. PLOS authors have the option to publish the peer review history of their article (what does this mean?). If published, this will include your full peer review and any attached files.

Reviewer #1: No

Reviewer #2: **Yes: **Zhi-Gang Shen

---

## [Author Response · Author response to Decision Letter 0]

8 Jan 2021

General comment

While revising the data to answer the reviewers’ comments, we identified 4 fish with clear inconsistent BL and BW measurements at 103 and 110 dph. These fish did not have out of range measurement within a given date, and this is why we did not identify them previously, but their measurements between dates were not compatible. We removed these data from the analysis, resulting in slightly different means and SD for BW, SL and DGC at 103 and 110 dph. 

Reviewer #1: Major comments

1. Daily growth rates were estimated only for weight (and not for SL) by using a formula, which in not very common in the literature (instead e.g. of the formula based on the exponential growth model, (e.g. lnW2-lnW1/Dt)). As weight was directly measured on individual fish only after the 6th sampling (117 dph), DGR estimations for the earlier period (83-110 dph) were based on back-calculated weight estimations (based on estimated morphometric relationships). A DGR estimation on the primarily measured SL would better support the results of the ms on the higher growth rate of females at 96-103 dph period.

Response: DGC was preferred to SGR because it has been demonstrated to be more stable than SGR over the life of the fish and to linearly increase with temperature (Bureau et al, 2000; https://www.aquatech.com.ve/pdf/bureau.pdf). This makes possible the comparison between data from different periods of fish life, although even with DGC, different growth stanzas may exist (Dumas et al., 2007. Aquaculture 267, 139 146).

DGC is not appropriate for SL. If fish do not change shape (Fulton’s K), the stability of DGC implies that growth in length (which is proportional to BW1/3) is linear according to temperature. Raising BL to ^(1/3) would create links between the residuals of the regression and age.

2. In many sampling points, body size was significantly larger in females than in males (Table 1). Despite this result, sex-related differences in DGR were significant only in one time period (96-103 dph, Table 2). The lack of agreement in these two approaches weakens the value of the findings. Especially because significant DGR differences were observed for a period where sample sizes were relatively small and not the same between the start and end points (96 and 103 dph respectively, Table 1, 2).

Response: Since DGC estimation implies the starting and the final weight, for each age interval we considered the animals that had both initial and starting weight available. In the table, we reported the real number of animals for which the DGC estimation was possible. Since DGC is a growth rate, similar growth rates would result in the persistence of differences between males and females if they exist at the beginning of the period. Thus, we do not see this as a lack of agreement between the two methods.

3. Given the availability of size-age data for a series of 15 sampling points, the estimation of growth curves on the basis of Ricker growth model (size=a*exp(r*age)) could solve the issue of comment No2, by involving a better statistical approach, with more accurate growth rate estimates. As during the studied period, different growth stanzas may exist, authors could use a) a piecewise linear regression to identify the inflexion points in the growth curves, and b) ANCOVA tests to compare the specific growth rate and “a” parameter between the two sexes (for each growth period separately). Such an approach is given in the figure below for the first growth period (based on the mean data provided in Table 1). The period of significant DGR differences is indicated by the circle. Arrow indicates a point of comparatively large residual value, which might partially explain the observed differences in the DGR for the period of 96-103 dph.

Response: We built the growth curves based on Ricker growth model for SL and BW, but we observed that this model seems to be not appropriate, as residuals were not even along the timeline. We instead built regressions of BW1/3 and untransformed SL on degree days above 10 °C, which is also coherent with the DGC approach. Doing this will yield similar results to the Ricker growth curves, but with a better fit.

4. Growth models should be estimated for both length and weight. Also, I would suggest the authors to test sex-related differences in fish condition factor (e.g. Fulton’s K).

Response: Thanks for the comment. We estimated Fulton’s K condition factor and we performed statistical analyses to test sex-related differences. It is an added value to the paper, since it is the first time that Fulton’s K has been estimated in such early life stages of European sea bass (line 164-166).

5. Table S2. Fish volume was estimated as a function of length and depth (height, line 118). Therefore, its simultaneous use with length and height as independent variables in the multiple regression models (with W as dependent variable) statistically is not recommended.

Response: Agree. We now use a model where height, length and volume were not simultaneously used, and we rearranged everything according to this change (line 192).

6. Fish size and not age should be primarily used as indicative of ontogenetic progress, as well as for the comparison of the min fish size where SDD is reported by the different studies.

Response: Agree. We have always indicated both size and age (the first indication is the size, followed by the age), because we prefer to maintain the link with the tables (where the age is always indicated); in some cases, we also indicated the degree days.

Reviewer #2: General comments

The work entitled “Sex dimorphism in European sea bass (Dicentrarchus labrax L.): new insights into sex-related growth patterns during very early life stages” studied sexual size dimorphism in European sea bass. There are three main concerns about the MS:

(1) The main weakness of the MS is lacking of histological observation of sex differentiation progress. Sex differentiation in fish is related to age (days post-fertilization), rearing conditions (e.g. temperature, nutrition), growth rate, and body weight/length. Without gonadal histology will make the results to be uncertain.

Response: The sexing has been performed at the end of the experiment (358 dph or 3229 degree days above 10 °C), when the identification through direct macroscopic observation of the gonads is possible and effective. 

The study was not focused exactly on sex differentiation progress, but on the growth patterns of a batch of fish that were individually followed from a very early stage and, only at the end, recognized as males or females. Anyway, it is a very good point to discuss in a future perspective section and to implement in a new experiment.

(2) There are usually three aspects of sex differentiation, cellular, morphological, and molecular sex differentiation. Molecular sex differentiation is much earlier than cellular and morphological sex differentiation, and is the cause of the other two. However, it is usually difficult to detect molecular sex differentiation until a late stage, because of the sampling problem of undifferentiated or differentiating gonad. Therefore, without the detect of molecular sex differentiation or morphological sex differentiation, the main conclusion “early growth may be a cause rather than a consequence of sex determination in sea bass” is unreliable.

Response: We proposed this as a hypothesis or a suggestion; the sentence was then modulated according to the reviewer’s comment (line 29).

(3) Even though authors make progress in the SSD research using microchip tagging, the tagging process is also the main affecting factor of the results as authors discussed. Eliminating small sized fish during sorting for microchip tagging artificially altered the sex ratio of the experimental population. The sorting process may artificially advance and speed up the SSD because fast-growing females were dominating the population. In addition, as observed in the work, eliminating the most slow-growing male repressed the SSD. It may be the cause why the SSD was disappear in the late stage which is in contrast with previous reports.

Response: That is a possibility, and we have tried to discuss that. For the sake of clarity, we would like to underline that fish were not sorted before microchip tagging, fish were randomly chosen from a common garden tank and tagged (please see Faggion et al. 2020, doi: 10.1016/j.aquaculture.2020.734945). What we were discussing in the current paper was the fact that larvae were subjected to a “strong” manipulation (i.e. the microchip tagging), which might have acted itself as a sorting event, favoring stronger fish (that are likely to develop as females). Anyway, we have data from an “untagged” group, that was the control group of the previous work (see Faggion et al. 2020). In this previous paper, we have already reported that microchip tagging did not affect survival rates nor growth rates of the tagged group in comparison with the untagged group. 

Looking at the sex ratio of the two groups (sex was recorded in all the individuals involved at the end of the experiment), this is the situation in the 2 groups:

- 65.6 % females : 34.4 % males in the tagged group, with a proportion of 1.91 females per male;

- 54.9 % females : 45.1 % males in the control “untagged” group, with a proportion of 1.22 females per male.

We did not find any significant differences in the proportion of males and females between groups (χ2 = 1.96, p-value = 0.16).

We rewrote this paragraph in the Discussion section (lines 347-353).

Please also find some specific comments as follows.

Specific comments

1. Line 28: The term “sex determination” is misused here and thereafter in the following context.

Response: In European sea bass, sex determination does not happen at hatching but at some time after hatching, and it is both environmentally and genetically determined. Whether final sex depends on events that determine it before differentiation or to events happening during the initial phases of differentiation is not clear (see Vandeputte et al, 2020, https://onlinelibrary.wiley.com/doi/full/10.1002/ece3.6972). In this paper, we collected individual growth data in a batch of fish, and we used the final sex status to reconstruct the growth patterns of individuals that finally differentiated as males or females. When sex was recorded, it was already determined (as individuals were sexed at a stage of advanced juveniles), so we were not able to relate individuals to the sex differentiation process but only to the final sex status, hence the use of the term “sex determination”.

2. It should be SSD, not SDD.

Response: True, it was a typo, we correct all the mistakes.

3. Are there replicates for the experiments? It had not mentioned in the MS.

Response: The replicates are the individuals. In this work we were not testing different treatments, we have just performed individual observations of fish and all the fish were reared in the same environment, under the same conditions.

4. Discussion section. The citation about an unpublished work again and again is not appropriate.

Response: We provided the correct citation since the work has now been published (see citation #26).

---

## [Decision Letter · Decision Letter 1]

16 Feb 2021

PONE-D-20-28397R1

Sex dimorphism in European sea bass (Dicentrarchus labrax L.): new insights into sex-related growth patterns during very early life stages

PLOS ONE

Dear Dr. Allal,

Thank you for revising your manuscript. After careful consideration, we feel that it has merit but does not fully meet PLOS ONE’s publication criteria as it currently stands. Therefore, we invite you to submit another revised version of the manuscript that addresses the points raised by reviewer 2.

We look forward to receiving your revised manuscript.

Kind regards,

Hanping Wang

Academic Editor

PLOS ONE

Reviewers' comments:

Reviewer's Responses to Questions

**Comments to the Author**

1. If the authors have adequately addressed your comments raised in a previous round of review and you feel that this manuscript is now acceptable for publication, you may indicate that here to bypass the “Comments to the Author” section, enter your conflict of interest statement in the “Confidential to Editor” section, and submit your "Accept" recommendation.

Reviewer #1: All comments have been addressed

Reviewer #2: (No Response)

2. Is the manuscript technically sound, and do the data support the conclusions?

Reviewer #1: Yes

Reviewer #2: No

3. Has the statistical analysis been performed appropriately and rigorously? 

Reviewer #1: Yes

Reviewer #2: I Don't Know

4. Have the authors made all data underlying the findings in their manuscript fully available?

Reviewer #1: Yes

Reviewer #2: Yes

5. Is the manuscript presented in an intelligible fashion and written in standard English?

Reviewer #1: Yes

Reviewer #2: Yes

6. Review Comments to the Author

Reviewer #1: The authors have adequately responded to my comments. Given the existence of 5-6 previous papers on the topic (for the given species, the most of which have been already cited and discussed by the ms), I would recommend to the authors to include a line in the abstract stating that their results verify the results of previous studies

Reviewer #2: The authors had answered all my three inquiries but had revised nothing in the new MS version.

The most important conclusion is “early growth may be a cause rather than a consequence of sex determination in sea bass”. The authors stated that “The study was not focused exactly on sex differentiation progress”, but the conclusion is closely related to sex differentiation, or so-called “sex determination”. I do not see any results that will support this conclusion. Sex differentiation is a process. For European sea bass, it is a relatively long process. As illustrated in Vandeputte et al. (Eco & Evo, 2020), sex differentiation initiated as early as 33 DPH, which is earlier than sexual size dimorphism in the present work. Why authors draw opposite conclusions for the same group of authors?

Besides, the authors used the term “sex determination” to infer when sex is determined. However, the term “sex determination” is recognized as “what determines sex”, such as genetics or environments.

7. PLOS authors have the option to publish the peer review history of their article (what does this mean?). If published, this will include your full peer review and any attached files.

Reviewer #1: No

Reviewer #2: **Yes: **Zhigang Shen

---

## [Author Response · Author response to Decision Letter 1]

10 Mar 2021

Reviewer #1: 

The authors have adequately responded to my comments. Given the existence of 5-6 previous papers on the topic (for the given species, the most of which have been already cited and discussed by the ms), I would recommend to the authors to include a line in the abstract stating that their results verify the results of previous studies.

>We have now included a sentence in the abstract to indicate that our results verify the results of previous studies (line 28).

Reviewer #2: The authors had answered all my three inquiries but had revised nothing in the new MS version.

>The first comment of the reviewer in the previous round was the following: “The main weakness of the MS is lacking of histological observation of sex differentiation progress. Sex differentiation in fish is related to age (days post-fertilization), rearing conditions (e.g. temperature, nutrition), growth rate, and body weight/length. Without gonadal histology will make the results to be uncertain”.

For this, we precised that sexing was performed at the end of the experiment, so that our results in terms of sex of the fish (as at that stage sexing is not ambiguous) are not uncertain. We did not feel we had to modify the text in this case. In any case, the main interest of the study is to follow fish growth individually, knowing their final sexual fate. It would not be feasible to have (lethal) gonadal histology and keep this individual tracking of growth performance.

The second comment of the referee was the following: “There are usually three aspects of sex differentiation, cellular, morphological, and molecular sex differentiation. Molecular sex differentiation is much earlier than cellular and morphological sex differentiation, and is the cause of the other two. However, it is usually difficult to detect molecular sex differentiation until a late stage, because of the sampling problem of undifferentiated or differentiating gonad. Therefore, without the detect of molecular sex differentiation or morphological sex differentiation, the main conclusion “early growth may be a cause rather than a consequence of sex determination in sea bass” is unreliable.”

> The full sentence was “This leads to the hypothesis that early growth may be a cause rather than a consequence of sex determination”. This is not a conclusion but clearly stated as an hypothesis. Even though, in the revised manuscript, we had replaced “may” by “might” to make the hypothetical nature of the statement even clearer. Even though this hypothesis is not proven, we feel important to formulate it as it may be the basis of future research.

The third comment of the referee was “Even though authors make progress in the SSD research using microchip tagging, the tagging process is also the main affecting factor of the results as authors discussed. Eliminating small sized fish during sorting for microchip tagging artificially altered the sex ratio of the experimental population. The sorting process may artificially advance and speed up the SSD because fast-growing females were dominating the population. In addition, as observed in the work, eliminating the most slow-growing male repressed the SSD. It may be the cause why the SSD was disappear in the late stage which is in contrast with previous reports.”

> For this point, we added a whole paragraph in the discussion, lines 353-360: “In a previous paper [17], it was reported that microchip tagging did not affect survival rates nor growth rates of the tagged group in comparison with the untagged group. Analyzing the sex ratio of the tagged group (i.e. the fish of the current experiment) and the untagged group (i.e. the control group in [17]), we observed 65.6 % of females and 34.4 % of males in the first case, with a proportion of 1.91 females per male; 54.9 % of females and 45.1 % of males in the second case, with a proportion of 1.22 females per male. The proportion of males and females was not statistically different between groups (χ2 = 1.96, p-value = 0.16), suggesting that the microchip tagging was not the cause of the unbalanced sex ratio we observed in our experiment.”

Considering those three points, we think it is not fair to say that we revised nothing in the manuscript, or maybe the referee got a wrong version of the manuscript.

The major criticism of the referee for the present version is the following:

The most important conclusion is “early growth may be a cause rather than a consequence of sex determination in sea bass”. The authors stated that “The study was not focused exactly on sex differentiation progress”, but the conclusion is closely related to sex differentiation, or so-called “sex determination”. I do not see any results that will support this conclusion. Sex differentiation is a process. For European sea bass, it is a relatively long process. As illustrated in Vandeputte et al. (Eco & Evo, 2020), sex differentiation initiated as early as 33 DPH, which is earlier than sexual size dimorphism in the present work. Why authors draw opposite conclusions for the same group of authors?

Besides, the authors used the term “sex determination” to infer when sex is determined. However, the term “sex determination” is recognized as “what determines sex”, such as genetics or environments.

> First, as already stated before, “This leads to the hypothesis that early growth may (indeed now “might”) be a cause rather than a consequence of sex determination” is not a conclusion but an hypothesis which is compatible with the data in the paper and which we find interesting to highlight for future research. Please note that we also propose an alternative hypothesis, that differentiation between males and females starts earlier than what has been evidenced until now (line 322).

We feel that one of the key points of misunderstanding with the referee lies in the use of the words “sex determination” and “sex differentiation”. Our theoretical frame is that used, for example, by Devlin and Nagahama (Aquaculture, 208: 191-364, 2002) who mention “assignment of gonad determination as well as subsequent differentiation”. In this framework, which is generally accepted, vertebrates start their life with an undifferentiated gonad, which differentiates as male or female. If sex determination is purely genetic, sex is determined at conception, and there is no question of timing. If sex determination is environmental (or both environmental and genetic, as in sea bass), then timing is critical, and sex determination only happens once the individuals have received and integrated environmental (and eventually genetic) cues that direct them either towards male or towards female sex differentiation. This timing of sex determination, stricly speaking, is when differentiation is engaged towards one sex and cannot be reversed to the other sex. In the sea bass, it is widely recognized that sex is fully determined at ~80 mm body length, where morphological differentiation of the gonads becomes visible and sex cannot be changed anymore by temperature or administration of exogenous stroids (Piferrer et al., 2005, doi:10.1016/j.ygcen.2005.02.011). What we see in Vandeputte et al. (Ecol. Evol., 2020), is that indeed sex is not determined before that, as sex-ratio can be modified by temperature treatments until 230-244 days at 16°C, an age at which fish reach 75-80 mm. 

Interestingly, molecular differentiation can be seen before this stage, although sex remains labile (thus not determined), as highlighted just before. Blazquez et al (2009 DOI: 10.1002/jez.b.21286) found that at ~50 mm body length, differential expression of cyp19a1a could be seen between future males and future females. 

In the paper by Vandeputte et al. (Ecol. Evol., 2020), we did not demonstrate that “sex differentiation initiated as early as 33 dph”, as stated by the referee. We showed that temperature at 32 dph did have an impact on sex determination, and thus on sex differentiation at a much later time. More specifically, all treatments (control, switch at 32, 53, 74 dph) had similar expression levels of cyp19a1a and gsdf at 75 dph, and differences appeared only at 96 dph. This shows that expression differences can occur at 96 dph (but not 75 dph). However; although molecular differentiation seems to have started at 96 dph, sex is not yet determined, as is shown by the fact that the same control fish (reared at 16°C) which exhibit a higher expresssion of gsdf at 96 dph can end up 29% female when switched to 21°C at 104 dph or only 8.6% female when switched to 21°C at 230 dph. Thus, although molecular differentiation may appear around 96 dph, sex is not determined before a body length of 75-80 mm is reached. Our interpretation is thus that this molecular differentiation, which happens during the labile period, is not necessarily indicative of a definitive sexual fate of the gonad, but more of the integration of environmental and genetic cues affecting sex determinantion. Growth could play a role here especially through what has been proposed as the size threshold model for sex determination, proposed by Ospina -Alvarez and Piferrer (2008, http://dx.doi.org/10.1371/journal.pone.0002837), under which “when a critical time is reached during development, a sexually undifferentiated gonad will develop as an ovary or as a testis depending on whether it has attained a certain size above or below a threshold”.

We detailed this at lines 328-336, and again, please read this as a hypothesis, not as a conclusion. This is further highlighted at lines 336-339.

---

## [Editor Report · Decision Letter 2]

5 Apr 2021

Sex dimorphism in European sea bass (Dicentrarchus labrax L.): new insights into sex-related growth patterns during very early life stages

PONE-D-20-28397R2

Dear Dr. Allal,

We’re pleased to inform you that your manuscript has been judged scientifically suitable for publication and will be formally accepted for publication once it meets all outstanding technical requirements.

Kind regards,

Hanping Wang

Academic Editor

PLOS ONE

---

## [Editor Report · Acceptance letter]

12 Apr 2021

PONE-D-20-28397R2 

Sex dimorphism in European sea bass (*Dicentrarchus labrax* L.): new insights into sex-related growth patterns during very early life stages 

Dear Dr. Allal:

I'm pleased to inform you that your manuscript has been deemed suitable for publication in PLOS ONE. Congratulations! Your manuscript is now with our production department. 

Kind regards, 

on behalf of

Dr. Hanping Wang 

Academic Editor

PLOS ONE